# Towards a More Efficient In and Ex Situ Conservation of Sri Lankan Wild Rice Species

**DOI:** 10.3390/plants12112149

**Published:** 2023-05-29

**Authors:** Thasajini Sajeevan, Andrea Mondoni, Malaka Wijayasinghe, Gehan Jayasuriya, Minindu Kumarage, Simone Orsenigo

**Affiliations:** 1Department of Earth and Environmental Sciences, University of Pavia, 27100 Pavia, Italy; 2Department of Biological Sciences, Faculty of Applied Sciences, South Eastern University of Sri Lanka, Oluvil 32360, Sri Lanka; 3Department of Biological Sciences, Faculty of Applied Sciences, Rajarata University of Sri Lanka, Mihintale 50300, Sri Lanka; 4Department of Botany, Faculty of Science, University of Peradeniya, Peradeniya 20400, Sri Lanka; 5Department of Geology, Kansas State University, Manhattan, KS 66506, USA

**Keywords:** conservation, dormancy, germination, seed, wild rice

## Abstract

Five species of wild *Oryza* (*O. nivara*, *O. rufipogon*, *O. eichengeri*, *O. rhizomatis* and *O. granulata*), including the endemic species *O. rhizomatis*, have been recorded in Sri Lanka. These species are facing continuous decline in their populations due to natural and anthropogenic processes, with habitat loss being the main threat. This study aimed to provide information on the distribution, the current status of ex situ and in situ conservation, and to identify high-priority species and sites of wild rice in Sri Lanka, in order to improve the effectiveness of conservation efforts. Occurrence records of Sri Lankan wild rice species were collected from literature, gene banks, and field surveys. The distribution of these species was mapped, and areas with high species richness were identified. A gap analysis was conducted to determine the high-priority areas and species for ex situ and in situ conservation. It was found that about 23% of the wild rice populations in Sri Lanka were within protected areas, and by expanding these protected areas by 1 km, an additional 22% of the populations located on the border of these areas could be effectively conserved. Our analysis also revealed that 62% of Sri Lankan wild rice populations were not represented in gene banks. The species-rich areas were found to be in only two districts (Polonnaruwa and Monaragala), and less than 50% of these areas were within protected areas. Based on these findings, *O. rhizomatis*, *O. eichengeri*, and *O. rufipogon* were identified as high-priority species for in situ conservation. Ex situ collections were also deemed necessary for *O. granulata* and *O. rhizomatis* to ensure diversity representation in gene banks.

## 1. Introduction

Crop Wild Relative (CWR) is a “wild plant taxon that has an indirect use derived from its relatively close genetic relationship to a crop” [1]. Approximately 645 species of CWRs have been reported in Sri Lanka [2], a biodiversity hotspot (considered together with the Western Ghats of India [3]). Among these, Sri Lanka hosts five CWRs of rice, namely, *Oryza eichingeri* Peter, *Oryza granulata* Nees and Arn. ex Watt, *Oryza nivara* S.D. Sharma & Shastry, *Oryza rhizomatis* D.A.Vaughan, and *Oryza rufipogon* Griff. *O. rhizomatis* is a wild rice species endemic to Sri Lanka [4], while *O. nivara* and *O. granulata* are native to Southeast Asia. Moreover, the common wild rice species, *Oryza rufipogon* is the putative ancestor of the Asian cultivated rice (*O. sativa*) [5] and *O. eichengeri* is distributed in Central Africa and Sri Lanka.

Wild *Oryza* species have been considered as sources of novel alleles in rice breeding programs [6]. They have a wide tolerance capability against both biotic and abiotic stresses. Specifically, wild rice species contain alleles that allow higher adaptability to arid climatic conditions when compared to commercial genotypes [7,8]. Wild rice species possess many beneficial agronomic traits, such as resistance to diseases, insects, pests, drought, salt, alkali, and high temperature. These traits include resistance to bacterial blight in *O. rufipogon* and *O. nivara*, resistance to blast in *O. rhizomatis*, and high biomass in *O. rufipogon* [9,10]. *O. rufipogon* has been used as a source of genes to improve the elongation ability [11], high tolerance for salinity [12], and low-temperature tolerance [13]. Moreover, all five Sri Lankan wild rice species have been utilized for developing brown leaf hopper-resistant rice varieties through breeding [10]. These are considered one of the major pests in Sri Lankan rice fields. Nevertheless, as with many other CWRs throughout the world, wild rice species in Sri Lanka have shown a rapid decline over the last few decades due to natural and anthropogenic processes [14,15]. Thus, immediate action should be taken to conserve these important species. To this end, identifying and prioritizing wild rice species and areas to be conserved are essential, preliminary steps toward setting effective in and ex situ conservation practices [16,17]. Identification of high-priority species and priority areas for conservation is crucial when planning effective conservation strategies and for policy development.

Gap analysis has been used in many studies to identify high-priority areas and species for in situ and ex situ conservation actions [17]. Burley (1988) first described the concept for the identification of “conservation gaps” as a process to identify and classify the various elements of biodiversity and examine the existing system of protected areas [18]. In Sri Lanka, protected areas are defined as clearly demarcated geographical spaces managed by the Departments of Forest Conservation and Wildlife Conservation, aimed at achieving the long-term conservation of nature and associated ecosystem services and cultural values [19]. These protected areas include Strict Nature Reserves, Forest Reserves, Forest Corridors, National Parks, and Sanctuaries, and are legally protected to varying extents to prevent encroachment and exploitation of forest resources [20]. They play a crucial role in safeguarding Sri Lanka’s natural heritage and are managed through legal or other effective means to ensure their conservation. Thus, wild species in a protected area are more protected from anthropogenic threats. The goal of gap analysis is to ensure that all ecosystems and areas rich in species diversity are represented adequately in biodiversity management areas. Areas identified as important via gap analysis can then be examined more closely for their biological qualities and management needs [21]. The level of biodiversity outside the protected areas system needs to be taken into account in future national conservation planning, particularly for CWR conservation. As an example, Necla et al. (2019) have studied the conservation gaps of CWRs in Turkey [22]. They focused on finding species-rich areas, ex situ representation of CWR taxa, and specifying the highest priority in situ locations. A similar study was conducted by Medeiros et al. (2021) to fill the gaps in germplasm collections of Brazilian CWR [23]. By comparing the range of natural diversity with the diversity already conserved ex situ in gene banks or in situ in natural reserves, the authors provided recommendations for additional conservation actions. Likewise, Nduche et al. (2022) performed in situ and ex situ conservation gap analyses of West African priority crop wild relatives. They identified high-priority species and sites and filled the identified in situ and ex situ conservation gaps [19]. In a separate study, Linsky et al. (2022) addressed the global conservation gap analysis of *Magnolia* species [20]. This analysis highlighted the need to increase well-documented and genetically diverse ex situ collections, particularly for species in areas where in situ habitat loss and climatic threats are greatest. Moreover, Khaki et al. (2021) conducted a study on the in situ and ex situ conservation gap analyses of crop wild relatives from Malawi [21]. They analyzed the representativeness of the conserved ecogeographic diversity of target taxa in ex situ collections to identify ex situ conservation gaps and identified priority areas for ex situ collections.

Liyanage et al. (2002) conducted a survey and mapping of Sri Lankan wild rice species, describing their morphological characteristics, phenology, and microhabitats. They also mapped the distribution locations of these species in different agro-ecological zones. Vaughan (1990) clarified the confusing taxonomy of the Sri Lankan *Oryza officinalis* complex by recognizing a new rhizomatous species, *Oryza rhizomatis* [4]. Sandamal et al. (2018) provided important insights into the population genetics and evolutionary history of *O. nivara* and *O. rufipogon* populations in Sri Lanka, which has significant implications for in situ conservation and management of genetic resources [22]. Ratnayake et al. (2021) found that Sri Lankan wild *Oryza* species are under threat from future climate change projections and suggested that their conservation in existing habitats and ex situ conservation is crucial [12]. Sandamal et al. (2021) highlighted the importance of conserving a country-specific rice germplasm for developing climate-smart cultivars, which can be achieved by enhancing the diversity of cultivars using valuable genetic resources available in wild relatives through breeding [23]. A comprehensive collection of distribution data of wild rice species is available in the literature [14,15,24]. However, species distribution data are not updated and none of the studies identified high-priority species and areas for conservation of wild rice species. Therefore, the aim of this study was to provide information on distribution, high-priority species, and areas for effective in situ and ex situ conservation of wild rice species in Sri Lanka. The specific objectives include: (a) producing an updated distribution map of wild rice species in Sri Lanka, (b) investigating the proportion of their populations located inside and outside existing protected areas, and (c) estimating the number of ex situ seed collections currently stored in germplasm banks. This information is used to identify new high-priority areas for in situ conservation and to determine species/populations that are not yet represented in ex situ seed bank. 

## 2. Results

### 2.1. Distribution Map of Study Species

GPS coordinates of 72 seed accessions (10 for *O. eichengeri*, 2 for *O. granulata*, 29 for *O. nivara*, 20 for *O. rhizomatis,* and 11 for *O. rufipogon*) were collected from the IRRI database (Appendix A). A total of 225 geographical distribution data of 225 populations were collected from the literature. 

IRRI (Philippine) and PGRC (Sri Lanka) were the only two places where seed accessions of wild rice species of Sri Lanka are stored (Appendix A). Rice Research Institute, Bathalagoda and Kew Millennium Seed Bank hosted no accessions of Sri Lankan wild rice species. PGRC collecting locations were not made available due to internal policy; therefore, subsequent gap analysis is based on GPS locations provided only by IRRI. The distribution pattern of Sri Lankan wild rice species varied across species (Figure 1). *O. nivara* is localized in dry and intermediate zones of Sri Lanka in Jaffna, Vavuniya, Puttalam, Anuradhapura, Kurunegala, Matale, Polonnaruwa, Badulla, Monaragala, Hambantota, Matara, and Kalutara districts. Eight populations of this species were recorded as extinct according to the field survey conducted during 2020–2022, but four new populations were recorded in the northern part of Sri Lanka (Figure 1). *O. rufipogon* grows in intermediate and wet zones in Gampaha, Kalutara, Matara, and Hambantota districts. *O. rhizomatis* is restricted to the dry zone of Sri Lanka in Anuradhapura, Puttalam, Kurunegala, Monaragala, Ampara and Hambantota districts. *O. granulata* and *O. eichengeri* were distributed in dry and intermediate zones. *O. granulata* is located in the Polonnaruwa, Monaragala, and Ratnapura districts, while *O. eichengeri* occurred in Anuradhapura, Matale, Polonnaruwa, Kandy, Monaragala, and Ratnapura districts. Field survey confirmed known populations of *O. granulata* and *O. eichengeri*.

### 2.2. Species-Rich Areas of Wild Rice in Sri Lanka

The wild rice species richness is variable across the grid cells (10 km × 10 km), ranging from zero to three. Overall, 82, 15 and 3% of the grid cells contained one, two and three species, respectively (Figure 2). There were no grid cells containing four or five species. *O. nivara*, *O. eichengeri,* and *O. rhizomatis* were the species that occur in the grid cell with three species in the Monaragala district, while *O. granulata*, *O. nivara,* and *O. eichengeri* occurred in the grid cells with three species in the Polonnaruwa district (North-Central dry zone) (Figure 2). 

### 2.3. Population Size of Sri Lankan Wild Rice Species

Sixty-eight percent of the populations visited during the field visits have <250 plants, only 12% of the *O. rhizomatis* populations have more than 1000 mature individuals, while 82% of the populations have <250 mature individuals. For *O. nivara,* 67% of the populations have <250 mature individuals, while 33% of the populations have <1000 mature individuals. Fifty five percent of *O. rufipogon* populations have <250 mature individuals, while 45% of the populations with <1000 mature individuals. In *O. granulata* and *O. eichengeri*, all the checked populations have <50 and <1000 mature individuals, respectively.

All populations of *O. nivara* located within the protected area had fewer than 50 individuals. For *O. rhizomatis*, 42% of the recorded populations inside the protected area had less than 50 individuals, while 15% had less than 250 individuals. During the field survey, all populations of *O. rufipogon*, *O. granulata*, and *O. eichengeri* were observed outside the protected areas (see Table 1). 

### 2.4. In Situ Conservation: Gap Analysis and Degree of Protection

Overall, 36 and 33% of the populations of *O. nivara* and *O. granulata* were inside the protected area, respectively. Considering *O. rhizomatis,* an endemic species, only 29% of populations were found within protected areas. Furthermore, only 27% of *O. eichengeri* populations occurred inside protected areas. On the contrary, none of the *O. rufipogon* populations were inside protected areas. Thus, *O. eichengeri*, *O. rhizomatis,* and *O. rufipogon* can be considered high-priority species for future in situ conservation, since <30% of these species were included in protected areas. 

Interestingly, a limited expansion of the existing network of protected areas by 1 km from the border will permit the incorporation of around 22% of the wild rice populations within the protected area. Considering grid cells interested by the presence of wild rice species, only 16% of the cells were effectively protected (Figure 3), while 36% were ineffectively protected and 49% were totally unprotected. Totally unprotected areas are present in Vavuniya, Anuradhapura, Puttalam, Gampaha, Ratnapura, Matara, Matale, Hambantota, Kurunegala, Kalutara, Monaragla, Trincomalee, and Ampara districts. Effectively protected areas are present in the Puttalam, Anuradhapura, Monaragala, and Hambantota districts of Sri Lanka. About 8% of the effectively protected areas were located in Gal oya national park, 33% in Wilpattu national park, and 58% in Yala national park. The two areas with the highest wild rice species richness, (in Polonnaruwa and Monaragala, respectively) are both ineffectively protected. 

### 2.5. Ex Situ Conservation: Collection of Seed Accession and Gap Analysis

IRRI (Philippine) and PGRC (Sri Lanka) were the only two places in which seed accessions of wild rice species of Sri Lanka were stored (Appendix A). The Rice Research Institute, Bathalagoda and Kew Millennium Seed Bank hosted no accessions of Sri Lankan wild rice species. PGRC collecting locations were not made available due to internal policy; therefore, subsequent gap analysis is based on GPS locations provided only by IRRI. Based on informal communication with the PGRC director (Sri Lanka), it has been confirmed that the IRRI and PGRC accessions are similar. IRRI hosted 42 populations of Sri Lankan wild rice species; among those were 61% and 47% of *O. rufipogon* and *O. eichengeri,* respectively. Moreover, there were 41% of *O. nivara* and less than 40% of the Sri Lankan populations of *O. granulata* and *O. rhizomatis* (33 and 25%, respectively). Consequently, *O. granulata* and *O. rhizomatis* can be considered high-priority species for ex situ conservation. Interestingly, most of the seeds in IRRI were collected between the 1970s and 1980s. 

## 3. Discussion

In this study, new populations were identified, extinction of previously reported populations was confirmed, and updated distribution maps of wild rice species in Sri Lanka were created using a combination of data from the literature, seed banks, and new field surveys. This is an important initial step towards the conservation and monitoring of existing populations of wild rice species, as the pre-existing information was outdated and dispersed. During the field surveys, the extinction of several populations of *O. rhizomatis* and *O. nivara* reported by Liyanage et al. (2002) and Liyanage (2010) was documented [2,24]. Notably, about one-third (28%) of the populations of the endemic *O. rhizomatis* previously reported in the literature no longer exist. Existing populations of *O. rhizomatis* were impacted by several threats, such as grazing by domestic animals and human activities related to agriculture, residential and commercial development, recreational activities, and the construction of transportation and service corridors. This alarming trend was previously noted by Pradeepa et al. (2017) [15], and it is likely to have worsened in the past decade, adding up to increasing negative impacts on habitat suitability due to climate change [20]. In contrast, all previously known populations of *O. rufipogon*, *O. granulata*, and *O. eichengeri* were confirmed during the field surveys conducted between 2020 and 2022, indicating inconsistent population dynamics across different *Oryza* species.

Generally, genetic diversity positively correlates with population size and influences population fitness [25,26]. The field visits conducted on wild rice populations revealed important insights into their status. More than 1/3 of the population in *O. rufipogon* and *O. nivara* have <50 individuals, and this % rises to 41 for *O. rhizomatis* and 50 for *O. granulata* In *O. rhizomatis,* 42% of the population with <50 individuals and 14% of the populations with <250 individuals were inside the protected area. The rest of the 58% of the *O. rhizomatis* population which were outside the protected area with <50 individuals were at higher risk of extinction. Meanwhile, only 12% of the *O. rhizomatis* populations had more than 1000 mature individuals, indicating rare populations with higher abundance. Prioritizing the conservation of these large populations is still crucial because they contain significant genetic diversity. In *O. nivara,* all the populations (100%) with <50 individuals are inside the protected area, so these populations are at less risk of extinction. On the other hand, the populations of *O. eichengeri* and *O. granulata* which were checked during the field visit had small population sizes and lacked both in situ and ex situ protection. Anyhow, these species occur in habitats that experience less disturbance from human pressure. These findings emphasize the need for continued monitoring and conservation efforts, including habitat protection, restoration, and addressing potential threats. Further research and management strategies are necessary to better understand the factors influencing wild rice population dynamics and abundance and ensure their long-term conservation. These findings provide valuable insights for the conservation and management of wild rice populations, emphasizing the importance of safeguarding this unique and ecologically important plant species for future generations. 

Species richness is a fundamental measurement of community and regional diversity that underlies ecological models and conservation strategies [27]. Our analysis revealed that the distribution of wild rice species richness in Sri Lanka is not uniform. Areas with high species richness were found to be situated in between protected and unprotected areas, with less than 50% of the grid cell being included in the former, resulting as ineffectively protected. For example, in the high species-rich area near Yala National Park (Figure 2), *O. rhizomatis* is located within the protected area, while *O. eichengeri* and *O. nivara* are located outside. Similarly, in the high species-rich area in the Polonnaruwa district, *O. nivara* is outside protected areas, while *O. granulata* and *O. eichengeri* are inside. In such cases, an effective conservation strategy might simply be to expand the existing protected areas. Enlarging protected areas has been shown to be an important and cost-effective way to protect threatened species, as it requires fewer resources than establishing new protected areas [28,29]. In southern Africa, conservationists have developed a bold initiative to dramatically enlarge and link many of the major protected areas by removing the boundary fences along national borders that separate many reserves [30]. In the East Usambara Mountains, the Tanzanian Government is attempting to enlarge and reconnect the nine largest blocks of forest in the region by means of wildlife corridors [31]. Along the same lines, several state governments, such as that of Goias, are focusing on creating protected areas and extending and consolidating existing protected areas, particularly with a view of establishing ecological corridors. Connectivity in terms of the protected area periphery seems more effective than corridors linking protected areas. Carlos et al. (2005) suggested that the conservation of the Brazilian Cerrado was achieved through strengthening and enlarging the system of protected areas and improving farming practices, and thus the livelihoods of local communities [32]. Creating ecological corridors and connecting protected areas was especially effective when connecting fragmented faunal populations. For plant species, enlarging the protected areas would be effective for including populations in the periphery of protected areas for more effective conservation. In our study, we found that by expanding the protected area by just 1 km, we can effectively protect an extra 22% of populations currently located outside protected areas.

According to our results, only 36 and 33% of the recorded populations of *O. nivara* and *O. granulata* are inside protected areas. Moreover, just 27 and 29% of populations of *O. eichengeri* and *O. rhizomatis* occur inside protected areas. Conversely, all populations of *O. rufipogon* were found outside protected areas. Consequently, the latter species should be considered a high priority for future in situ conservation actions, as protected areas in conservation retain more biodiversity than alternative land uses [33]. Nevertheless, an overall increment of in situ conservation of *Oryza* species is desirable, especially for some populations growing in unique and isolated microhabitats [4], which may show distinctive phenotypic characters and genetic diversity [34]. The establishment of new protected areas has already been practiced in Sri Lanka to conserve wild rice species. For example, Vanathavilluwa (in Puttalam district), the first in situ conservation site to conserve endemic *O. rhizomatis* was established under the Northwestern Province Environmental Statute (No: 12 of the 1990). This method has also been adopted for conserving threatened wild species in Sri Lanka, such as a freshwater swamp forest in Bulathsinhala, which was declared a protected area in 2007 [35] to conserve the critically endangered plant species *Stemonoporus moonii* (Thwaites) and *Mesua stylosa* (Thwaites) Kosterm. Similarly, Warathenna-Hakkinda protected area was declared in 2017 [35] to conserve several fauna and flora species, including four endangered *Cryptocoryne* (Araceae) species. In spite of this, the inclusion of all the wild rice populations in protected areas is not a viable option in Sri Lanka, as many populations are located close to or inside urbanized or agricultural areas. This pattern is reflected globally, with most of the biodiversity existing outside protected areas [36]. Most of the effectively protected areas are in the Wilpattu, Yala, and Galoya National Park. 

The gap analysis conducted to assess the level of ex situ conservation revealed that *O. granulata* and *O. rhizomatis* should be given higher priority, as only 33% and 25% of their populations are represented in seed storage for ex situ conservation. Ex situ conservation can have various goals, such as preventing species extinction, providing materials for plant translocation and habitat restoration, and enhancing crops [37]. The number of populations selected for seed storage is dependent on these goals. The primary objective of ex situ conservation of wild rice species is to aid in their conservation and preserve their genetic material for crop improvement. To protect these species from extinction, priority should be given to endangered populations, those that are found in diverse environmental regions, and those that possess high phenotypic differences [38]. In order to preserve genetic diversity for future crop breeding programs, it is recommended to establish ex situ collections based on available resources, representing as many populations of wild rice species as possible [37]. However, our analysis indicated that many of the known wild rice populations were not present as seed accessions at the International Rice Research Institute (IRRI) (62%; Figure 1, Appendix A). Moreover, several IRRI accessions are outdated, having been collected between 1964 and 1992. Gene bank accessions should be duplicated regionally and internationally to ensure effective and long-term ex situ conservation [39]. New collections are desirable, both for new and already stored populations, as plants regenerated from seeds stored for a long time (up to c. 60 years in this case) may no longer be able to adapt to current abiotic conditions and biotic interactions [40]. Additionally, seeds from newly discovered areas, such as *O. nivara* from the northern provinces, should be collected from their original sites as they grow in microclimatic conditions that differ from those available in the gene bank. Hence, it is necessary to plan new collections to increase the genetic variability of wild rice species in the future.

## 4. Materials and Methods

### 4.1. Distribution Map of Study Species

#### 4.1.1. Distribution Data from Literature Review ad IRRI Data Base

GPS coordinates of the wild rice species populations were obtained from recent scientific literature (from the last 20 years) and the International Rice Research Institute (IRRI). A total of 297 records derived from the IRRI database and the literature were georeferenced and organized in ArcGIS 10.3 (Esri, 380 New York CA 92373-8100).

#### 4.1.2. Distribution Data from Field Survey

Field surveys were conducted during 2020–2022 to determine the current situation of the populations (i.e., their presence, absence, and if present, whether the populations are threatened or not) reported in the abovementioned sources. For *O. nivara*, *O. granulata*, *O. rufipogon, O. eichengeri* and *O. rhizomatis* 53, 29, 48, 10 and 85% of the populations were checked for their presence among those reported in the recent literature and IRRI database (Appendix A). Moreover, 4 new localities were recorded during field surveys and added to the geodatabase.

GPS coordinates of populations gathered using the abovementioned methods were categorized as follows: “literature data”, coordinates of populations collected from the literature (i.e., from 2002 to present), and “IRRI data” coordinates of populations collected from the IRRI database. Further, if the population was found during the field survey, it was categorized as “existing”. If the population was no longer recorded during the field survey it was categorized as “extinct”. New localities were categorized as “new population”.

### 4.2. Population Size of Wild Rice in Sri Lanka

To determine the number of individuals in each population, two methods were used. If the populations comprised <100 mature individuals, the accurate number of mature individuals was counted. For big populations with >100 mature individuals, three quadrats with 1.5 m × 1.5 m of size were randomly located in the field and the number of individuals in each quadrate was counted. The average number of mature individuals per 1 m^−2^ was determined. Then, the total number of mature individuals in the population was estimated by multiplying the average number of individuals per m^2^ with the total area of the population. Finally, the population size was standardized according to the criterion C of the IUCN red list (i.e., <50, 51–250, 251–1000, >1000 mature individuals [41].

### 4.3. Species Rich Areas of Wild Rice in Sri Lanka

To identify the *Oryza* species-rich areas, geo-referenced data of the five studied species were merged in to a single map using ArcGIS software. Geographic data were upscaled to a 10 km × 10 km fixed grid, since the grid size is appropriate for revealing the species richness pattern at the national level, and this minimized sampling bias. Species richness was ranked from 1 to 5 according to the number of wild rice species present in a single cell. We defined species-rich areas as grid cells containing at least 3 *Oryza* species.

### 4.4. In Situ Conservation: Gap Analysis and Degree of Protection

To determine the gaps in in situ conservation, distribution maps of each species and species-rich areas were superimposed to the map of protected areas of Sri Lanka (http://protectedplanet.net/ accessed on 20 February 2022 [both under the Department of Wildlife Conservation and the Department of Forest Conservation]). The number of populations that fell within protected areas of each species was determined and the percentage of protected populations was calculated for each species. The protected area system was superimposed on the same 10 km×10 km cell grid layer. The protection level of wild rice species-rich areas was considered “effective” (highly protected areas) when ≥50% of the cell surface was included in a protected area, and “ineffective” (areas with little protection) when <50% of the cell surface was included in protected areas [42,43]. The cells located fully outside the protected areas were categorized as “totally unprotected”.

### 4.5. Ex Situ Conservation: Collection of Seed Accession and Gap Analysis

The availability of ex situ Sri Lankan wild rice seeds or plants was investigated in the following institute: IRRI (Philippine), Kew Millennium Seed Bank (UK), Plant Genetic Resources Center [PGRC] and Rice Research Institute Bathalagoda (Sri Lanka). 

Population locations that were taken from the IRRI database were compared against locations of the species recorded in the literature and the field survey. This way, populations stored in ex situ collections were identified and calculated as a percentage (populations stored ex situ/Total number of populations identified × 100) for each species. Priority species for ex situ conservation were identified according to the criteria proposed by Beyrouthy et al. (2019) [44]: high-priority species for ex situ conservation were the species with <40% of the populations stored in germplasm collections; medium-priority species were those with 40–70% of populations stored in germplasm collections; and low-priority species were those with 70–100% of populations stored in germplasm collections. 

## 5. Conclusions

*O. rhizomatis*, *O. eichengeri,* and *O. rufipogon* should be considered high-priority wild rice species for in situ conservation. Enlargement of existing protected areas or the institution of new small-sized protected areas in Hambantota, Puttalam, and Kalutara districts is suggested to increase the effectiveness of in situ conservation of high-priority species. Special attention should be given to the high-species rich areas, since these were categorized as ineffectively protected areas. Nearly half of the populations were outside protected areas and must be considered totally unprotected. Ex situ collections should be increased based on the ex situ gap analysis for *O. granulata* and *O. rhizomatis* since the IRRI and PGRC collections do not represent all the populations identified in recent literature and field surveys, and most of the accessions are outdated. 

## Figures and Tables

**Figure 1 plants-12-02149-f001:**
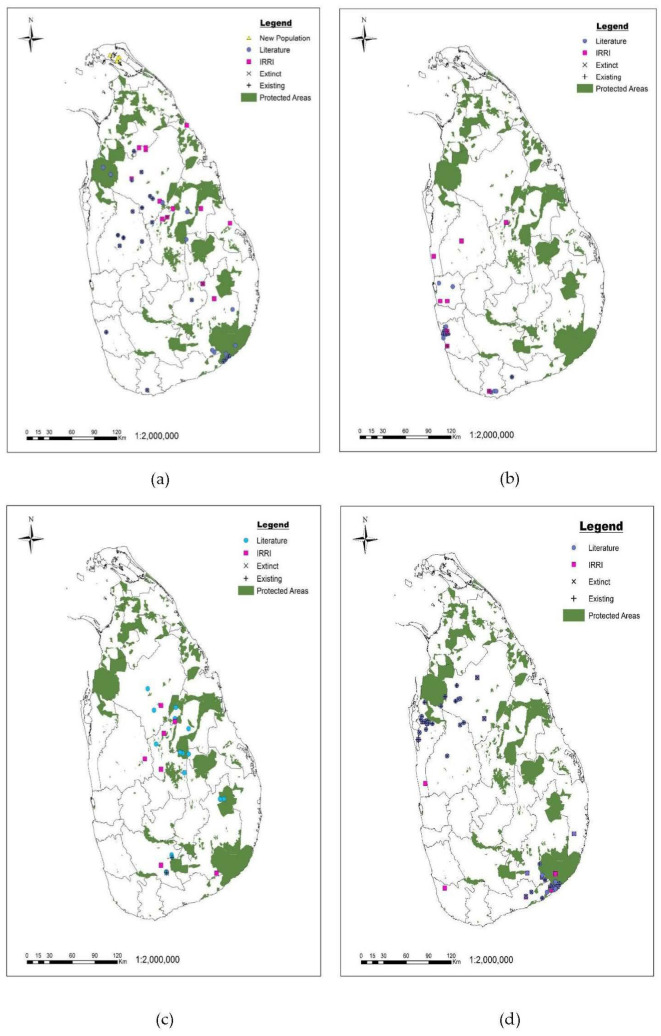
Distribution map of Sri Lankan wild rice species (**a**) *O. nivara*, (**b**) *O. rufipogon*, (**c**) *O. eichengeri,* (**d**) *O. rhizomatis,* (**e**) *O. granulata* based on the information from ex situ seed accessions held at IRRI (pink quadrats) and recent literature (blue circle) overlaid with the wildlife protected areas (under the Department of Wildlife Conservation and Department of Forest conservation of Sri Lanka). Confirmed (+) or extinct (×) or new populations (yellow triangle) found during the field survey are reported.

**Figure 2 plants-12-02149-f002:**
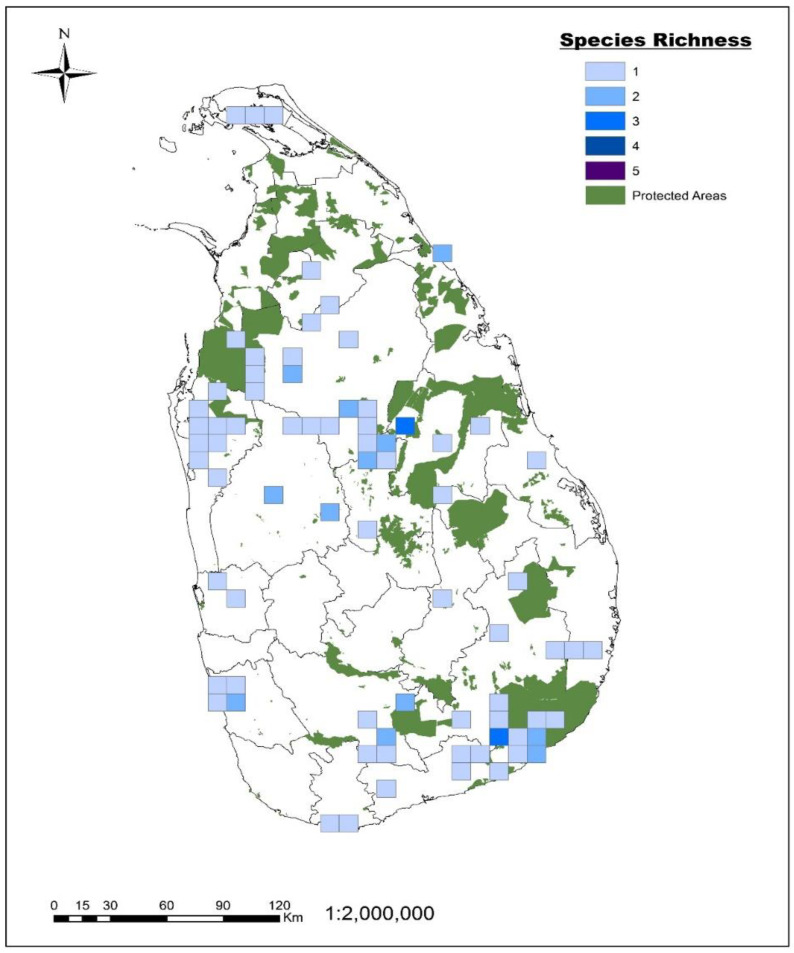
Species richness map of Sri Lankan wild rice species superimposed to protected areas (under the Department of Wildlife Conservation and Department of Forest Conservation of Sri Lanka). Colors indicate the number of wild rice species present in a 1 km × 1 km grid cell.

**Figure 3 plants-12-02149-f003:**
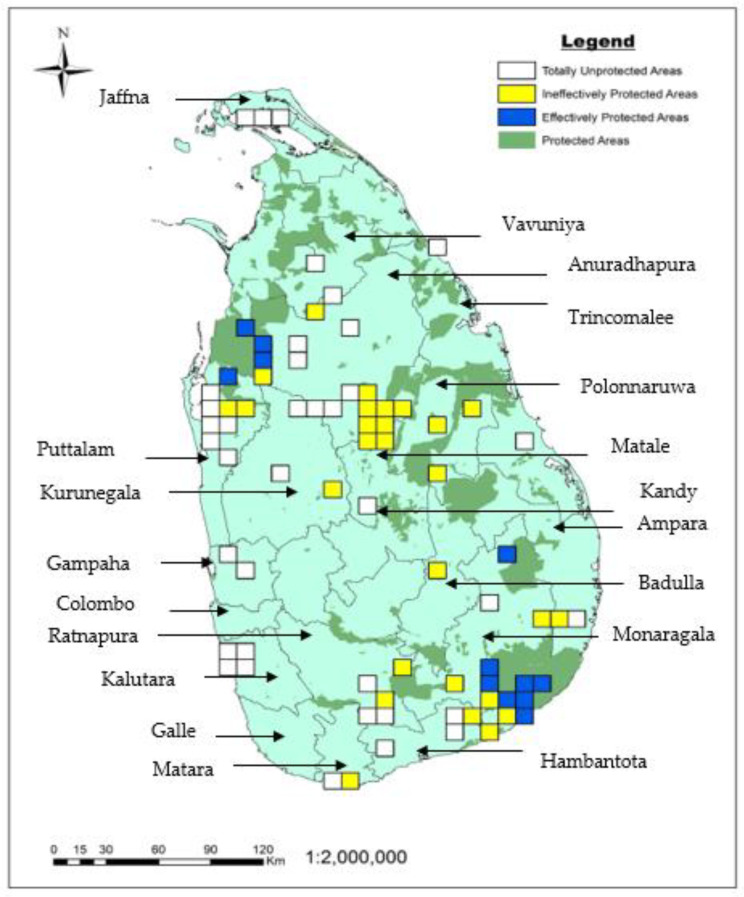
Distribution of Sri Lankan wild rice species over layered with the protected areas (green): blue squares (10 km × 10 km) represent effectively protected (>50% of cell surface included in the protected area); yellow squares (10 km × 10 km) represent ineffectively protected (<50% of cell surface included in the protected area); white squares represent totally unprotected cells.

**Table 1 plants-12-02149-t001:** Number of mature individuals and percentage of populations inside the protected area of Sri Lankan wild rice species sampled during the field visits (2020–2022) and ranked according to the criterion C of the IUCN red list (IUCN 2012).

Species (Total Number of Populations Visited)	Population Size (Number of Individual Plants in the Population)							
	<50		51–250		251–1000		>1000	
	Total%	Inside the PA%	Total%	Inside the PA%	Total%	Inside the PA%	Total%	Inside the PA%
*O. nivara* (12)	34%	100%	33%	0	33%	0	0	0
*O. rhizomatis* (17)	42%	41%	41%	14%	6%	0	12%	0
*O. rufipogon* (9)	33%	0	22%	0	45%	0	0	0
*O. granulata* (2)	100%	0	-	0	-	0	-	0
*O. eichengeri* (2)	50%	0	-	0	50%	0	-	0

PA—Protected Area.

## Data Availability

Not applicable.

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
