# Peer review of "Towards a More Efficient In and Ex Situ Conservation of Sri Lankan Wild Rice Species"

_plants, 2023, doi:10.3390/plants12112149_

Round 1
Reviewer 1 Report
The paper is an update on the distribution of wild Oryza species in Sri Lanka and on the degree of protection of which they benefit. It is well written and, even though it does not present any scientific novelty, it is interesting from the point of view of the practice of plant conservation. I would recommend it for publication after a minor revision.
Introduction
The introduction is too long and the paragraph on the Sri Lankan Oryza species (lines 88-100) is not well connected with the next one
Materials and Methods:
lines 304-305: Oryza rhizomatis results to be a synonim of O. eichengeri Peter that is not exclusive to Sri Lanka but native also to Africa
line 350: which gene bank database was used? If data were obtained by the databases of all four institutions listed above, please change “database” with “databases”
Results
Lines 177-178: Please, explain also in the text and not only in the caption of Figure 3, what is intended with “effectively protected”, “ineffectively protected” and “totally unprotected”.
Lines 193-197: these lines explain why only seed accessions from IRRI are plotted in Figure 1. I would advice therefore to move these lines at the beginning of the Results section, before introducing Figure 1
Discussion:
line 211: “no long exist” or “were no long existing” instead that “were no long exist”
Finally, I would recommend a general revision of the English language
Author Response
Introduction: The introduction is too long and the paragraph on the Sri Lankan Oryza species (lines 88-100) is not well connected with the next one
Reply: I have combined both paragraphs to make a connection
Materials and Methods: lines 304-305: Oryza rhizomatis results to be a synonim of O. eichengeri Peter that is not exclusive to Sri Lanka but native also to Africa
Reply: I have removed this paragraph from the materials and method since it is not appropriate to include there.
line 350: which gene bank database was used? If data were obtained by the databases of all four institutions listed above, please change “database” with “databases”
Reply: The population locations were taken from only IRRI data base
Results: Lines 177-178: Please, explain also in the text and not only in the caption of Figure 3, what is intended with “effectively protected”, “ineffectively protected” and “totally unprotected”.
Lines 193-197: these lines explain why only seed accessions from IRRI are plotted in Figure 1. I would advice therefore to move these lines at the beginning of the Results section, before introducing Figure 1
Reply: The explanation for ineffectively protected, effectively protected and totally unprotected areas were included in line 348-352.
Discussion: line 211: “no long exist” or “were no long existing” instead that “were no long exist”
Reply: I have changed it as ‘’were no longer existing’’

Reviewer 2 Report
General remarks
This study intend to update the distribution of populations of rice’s wild relatives (RWR) in Sri Lanka and implemented a gap analysis for in situ and ex situ conservation of RWR. While the general idea is commendable, the implementation of the study has at least three major shortcomings that reduce considerably its significance.
1-Regarding in situ conservation and gap analysis: Field surveys conducted during 2020-2022 to determine the current situation of the populations (i.e., presence, absence, and if presence whether the populations are threatened or not) covered a limited share of the populations reported in the literature and in IRRI database : “For O. nivara, O. granulata, O. rufipogon, O. eichengeri and O. rhizomatis 53, 29, 48, 10 and 85% of the populations were checked among those reported in the recent literature and IRRI database (Sup Tab 3)”.
2-Regarding ex situ conservation and gap analysis: due to lack of geographical position for populations maintained in Sri Lankan gene-banks, the analysis was limited to accessions present in IRRI gene-bank. And no connection was established between the accessions maintained in these two gene-bank.
3- Both in situ and ex situ updating and gap analysis are based on one “quantitative” data: presence versus absence. No information is available about the size of each population that constitutes a more accurate information on its vulnerability.
A more general issue limiting the significance of the study is the lack of information on the genetic diversity of the individual populations and genetic relationship between different populations of each species. This means some degree of characterization of each population using morphological descriptors or molecular markers.
Specific remarks: Please see my annotations on the manuscript.

Author Response
Reviewer 2
General remarks
This study intends to update the distribution of populations of rice’s wild relatives (RWR) in Sri Lanka and implemented a gap analysis for in situ and ex situ conservation of RWR. While the general idea is commendable, the implementation of the study has at least three major shortcomings that reduce considerably its significance.
1-Regarding in situ conservation and gap analysis: Field surveys conducted during 2020-2022 to determine the current situation of the populations (i.e., presence, absence, and if presence whether the populations are threatened or not) covered a limited share of the populations reported in the literature and in IRRI database : “For O. nivara, O. granulata, O. rufipogon, O. eichengeri and O. rhizomatis 53, 29, 48, 10 and 85% of the populations were checked among those reported in the recent literature and IRRI database (Sup Tab 3)”.
Reply: The limited number of populations were checked during the field survey for O. granulata, O. eichengeri (29 and 10% respectively) was due to their limited accessibility as these species are present in dense forest. Moreover, we decided to address sampling efforts more on the endemic species (e.g. O. rhizomatis) and/or species that tend to occur in more threatened or human impacted habitat (such as temporary ponds and swamps), assuming that species living in pristine habitats like dense forests should be less impacted by anthropogenic impact.
2-Regarding ex situ conservation and gap analysis: due to lack of geographical position for populations maintained in Sri Lankan gene-banks, the analysis was limited to accessions present in IRRI gene-bank. And no connection was established between the accessions maintained in these two gene-bank.
Reply: Based on informal communication with the PGRC director (Sri Lanka), it has been confirmed that the IRRI and PGRC accessions are similar and collections were done in the same period and have been duplicated in the two institutions
3- Both in situ and ex situ updating and gap analysis are based on one “quantitative” data: presence versus absence. No information is available about the size of each population that constitutes a more accurate information on its vulnerability.
Reply: I agree that adding information about the population size is worth. I have included data about the population size sampling/counting under materials and method (page 14, line 416-424), results (page 7, line 189-209) and discussion (page 11, line 281- 301). Large populations of O.rhizomatis with over 1000 mature individuals exist outside of protected areas, but their seeds are being conserved ex-situ. However, prioritizing the conservation of these large populations is still crucial because they contain significant genetic diversity. On the other hand, O.eichengeri and O.granulata have small population sizes and lack both in-situ and ex-situ protection, making their conservation a higher priority
4- A more general issue limiting the significance of the study is the lack of information on the genetic diversity of the individual populations and genetic relationship between different populations of each species. This means some degree of characterization of each population using morphological descriptors or molecular markers.
Reply: Despite we know that genetic diversity is extremely important to consider in conservation prioritization, this study was mainly focused on geographical and distributive data related to in-situ ad ex-situ conservation of Sri Lankan wild rice species. For sure we retain that in the future the knowledge of genetic diversity of single populations or genetic relationships between different populations of Sri Lankan wild rice species could be implemented and will be useful also for breeders and purposes that goes further conservation, but this goes beyond our research

Reviewer 3 Report
The study investigates the current distribution of the five wild rice species recorded in Sri Lanka. It reports that only a minority of existing populations live within protected areas and are represented in gene banks, thus highlighting the need for more effective protection measures. The study shows that expanding Sri Lanka protected areas by only one km would help protecting a further 23% of wild rice populations at a relatively low cost.
The paper is well organized and written in good English. My only observations are:
Line 102: indicate the dates of the last records. These are mentioned in the Discussion, but I think they should also be given in the Introduction.
Line 109: I’m not sure whether “will be” should be “is”
Line 149: something is missing
Figure 2: The legend should explain that the colours indicate the number of wild rice species present in the same site. Colours 4 and 5 might perhaps be omitted.
Lines 240-244: the two sentences appear to be incomplete.
A few minor suggestions have been incorporated in the pdf attached.

Author Response
The study investigates the current distribution of the five wild rice species recorded in Sri Lanka. It reports that only a minority of existing populations live within protected areas and are represented in gene banks, thus highlighting the need for more effective protection measures. The study shows that expanding Sri Lanka protected areas by only one km would help protecting a further 23% of wild rice populations at a relatively low cost.
The paper is well organized and written in good English. My only observations are:
Line 102: indicate the dates of the last records. These are mentioned in the Discussion, but I think they should also be given in the Introduction.
Reply: I have included it in lines 117- 119 -species distribution data are not updated and none of the studies identified high-priority species and areas for conservation of wild rice species.
Line 109: I’m not sure whether “will be” should be “is”
Reply: I have changed it as ‘’is’’
Line 149: something is missing
Reply: This is due to a text box overlapped with the figure legend. I have deleted it.
Figure 2: The legend should explain that the colours indicate the number of wild rice species present in the same site. Colours 4 and 5 might perhaps be omitted.
Reply: I have included that the colors indicate the number of wild rice species present in 1 km ´ 1 km grid cell. There are no grid cells with 4 or 5 species together in one grid cell. The maximum species richness is 3.
Lines 240-244: the two sentences appear to be incomplete.
Reply: I have made changes to make it complete.

Round 2
Reviewer 2 Report
Thank you for this feedback.
I read the authors' answers to my major remarks stated after my first review and reiterated after the second review. Unfortunately, these answers did not change my opinion about the overall mediocre scientific quality of the manuscript and its limited interest for the scientific community outside Sri Lanka.